# Development of Methods for Improving Flowering and Seed Set of Diverse Germplasm in Cassava Breeding

**DOI:** 10.3390/plants13030382

**Published:** 2024-01-27

**Authors:** Peter T. Hyde, Olayemisi Esan, Elohor Mercy Diebiru-Ojo, Peter Iluebbey, Peter A. Kulakow, Prasad Peteti, Tim L. Setter

**Affiliations:** 1Section of Soil and Crop Sciences, School of Integrative Plant Science, Cornell University, Ithaca, NY 14853, USA; 2Cassava Breeding Unit, International Institute of Tropical Agriculture, Ibadan 200001, Nigeria

**Keywords:** anti-ethylene, STS, cytokinin, floral development, seed set

## Abstract

Cassava breeding faces obstacles due to late flowering and poor flower and seed set. The acceleration of breeding processes and the reduction in each cycle’s duration hinge upon efficiently conducting crosses to yield ample progeny for subsequent cycles. Our primary objective was to identify methods that provide tools for cassava breeding programs, enabling them to consistently and rapidly generate offspring from a wide array of genotypes. In greenhouse trials, we examined the effects of the anti-ethylene silver thiosulfate (STS) and the cytokinin benzyladenine (BA). STS, administered via petiole infusion, and BA, applied as an apical spray, combined with the pruning of young branches, significantly augmented the number of flowers. Controls produced no flowers, whereas treatments with pruning plus either BA or STS alone produced an average maximum of 86 flowers per plant, and the combination of pruning, BA and STS yielded 168 flowers per plant. While STS had its primary effect on flower numbers, BA increased the fraction of female flowers from less than 20% to ≥87%, thus increasing the number of progeny from desired parents. Through field studies, we devised an optimal protocol that maintained acceptable levels of phytodamage ratings while substantially increasing seed production per plant compared to untreated plants. This protocol involves adjusting the dosage and timing of treatments to accommodate genotypic variations. As a result, cassava breeding programs can effectively leverage a diverse range of germplasm to develop cultivars with the desired traits.

## 1. Introduction

Cassava (*Manihot esculenta*, Crantz) is a warm-season crop grown throughout the tropics, primarily for its storage roots, which have a high starch content and serve as a staple food as well as a range of uses from high-value specialty starches to animal feed. Reliance on the global food chain for essential food commodities can be precarious when world events, such as the coronavirus pandemic and wars, disrupt global food supplies, laying bare the benefit of increased production of local crops. In Africa, cassava provides daily caloric needs to hundreds of millions of people [1,2]. New elite cultivars with novel diseases and agronomic traits are needed, but conventional cassava breeding and multiplication of planting stock is slow, taking upwards of twelve years [3].

Cassava breeding has gained considerable momentum due to an increase in financial resources devoted to the crop by national and international organizations over the last decade [4]. Numerous studies have focused on the utility and application of modern breeding methodologies, such as genomic selection and crop modeling [3,5], to improve breeding efficiency. Other important work is being aimed at identifying user trait preferences and developing methods of evaluating and quantifying processing traits such as firmness, mealiness and nutrient content [6,7], thus emphasizing the need to breed multiple varieties for a range of user needs. Additionally, breeding for resistance to insects such as greenmite [8] and whitefly and for diseases such as CBSD and CMV is critical for the durability of new cultivars [9,10].

All of these breeding approaches rely on genetic recombination and selection facilitated by cross pollination, which is a bottleneck in cassava breeding [5,11,12]. In cassava, a high fraction of genotypes are late flowering or never flowering at all, and the number of flowers ranges from profuse to sparse or none [13,14,15,16,17,18]. Efforts to speed up plant breeding by employing genomic selection and shortening the length of each breeding cycle [5,19] inevitably depend on the ability to hasten flowering and efficiently make crosses so that sufficient progenies are obtained for the next cycle.

Recently, progress has been made to identify factors that affect floral initiation and development in cassava. Cooler temperatures and longer photoperiods interact with the FT flowering regulatory system to influence the transition of the apical meristem to an inflorescence [14,16,17,20,21]. Floral development after initiation is affected by two plant hormones in cassava: ethylene and cytokinins. Ethylene negatively impacts floral development by causing young inflorescences to abort before reaching maturity [22,23]. Cytokinin alters cassava floral development so that a greater proportion are female [15,23]. Another factor that affects flower development is the extent to which shoot branches subtending floral primordia develop just below the terminal meristem. Pruning these fork branches after floral initiation prevents arrested inflorescence development and increases the number of flowers [15,23]. This work has been valuable in determining the fundamental biology of the system using controlled environments and a few model genotypes to develop an understanding of the biology of cassava. However, most of these studies focused on flower development and not seed development. Furthermore, the needed development of procedures effective on a wide variety of genotypes and growth conditions has not yet been performed.

In this paper, we elucidate, systematically develop and test a robust protocol to increase seed production in cassava crossing nurseries. This information indicates that treatments of anti-ethylene, cytokinin and pruning have individual and combined effects to improve flowering and seed production in cassava. Understanding the individual effects of these treatments is useful for future refinement of the process by breeders. We also report the validation of our optimized protocol by implementing and fine-tuning these methods for three years in practical field use on a set of 19 genotypes. This information will enable breeders to further refine treatment procedures for their specific germplasm and environments. 

## 2. Results

### 2.1. Greenhouse Trials of Plant Growth Regulators and Pruning Treatments

For greenhouse studies, we used a set of three standard reference lines used in breeding programs in Africa. Spray applications of the anti-ethylene plant growth regulator silver thiosulfate (STS), and the cytokinin benzyl-adenine (BA) were each effective in significantly (*p* ≤ 0.05) increasing the number of flowers produced on each inflorescence; however, combining the two plant growth regulators (PGRs) together stimulated a substantially greater number of flowers than either of the individual treatments (Figure 1, Experiment 1). There were no significant effects of the treatments on the days of flower appearance (Appendix A). Based on findings from a recent study from our lab [23], we tested an alternative method of hormone application whereby STS solution enters the xylem via a cut petiole. Applying STS using the petiole method was effective at significantly increasing the number of flowers produced; STS dosages of 0.125 mM to 1.0 mM all significantly increased the number of flowers compared to the control, though benefit plateaued between 0.25 mM and 1.0 mM STS (Figure 1, Experiment 2). Evaluating different combinations of BA and STS application methods (spray and petiole) indicated that applying STS via the petiole and BA as a spray to the apical region was the most effective combination, such that it produced approximately five times more flowers than the other combinations (Figure 1, Experiment 3). Petiole feeding involved small volumes of solution, and STS phytodamage was less than with spray, so STS concentration was boosted to 0.5 mM with petiole application. While controls produced no viable flowers, pruning the lateral branches during the early stages of inflorescence development modestly increased the number of flowers compared to the control, yielding an average maximum of 23 flowers per plant (Figure 1, Experiment 4). Treatments with pruning plus either BA or STS alone further increased the average maximum to 86 flowers per plant, and the combination of all three treatments, pruning, BA and STS, yielded 168 flowers per plant. 

The greenhouse experiments showed that beyond increasing the number of flowers, the BA treatments increased the percentage of female flowers. Treatments that included BA spray consistently produced more than 80 percent female flowers, significantly greater than the approximately 10 percent seen in flowers without BA spray (Table 1). Most notable was when BA was in combination with other treatments that increased the total number of flowers, such as the combined treatment of prune, STS and BA in Experiment 4; it produced not only a large number of flowers but also mostly female flowers.

### 2.2. Field Trials of Flowering in Response to PGR and Pruning Treatments

#### 2.2.1. Ibadan Location Field Trial of PGRs

Preliminary field trials indicated that to have a discernible effect, concentrations of STS in the field needed to be higher than those used in the greenhouse [23]. The field trials located in Ibadan involved spraying STS at 0.5 mM to the apical region. This was four times the minimum effective rate for the same compound when applied as a spray in the greenhouse (Figure 1, Experiments 1 and 2). Field trials in Ibadan with three standard reference lines that are used in Nigerian breeding programs indicated that spray applications of STS, BA and the combination of the two significantly increased the number of male and female flowers and increased the duration of flower production on a plant (flowering integral) (Table 2). The largest increase in flower numbers was with the combination of STS and BA. 

#### 2.2.2. Ubiaja Location Field Trials of PGRs and Pruning

Investigations over three years on a set of 19 genotypes representing lines actively used in the cassava breeding program with a range of diversity for flowering showed that the treatment combination of pruning, STS and BA increased the number of seeds produced on the first tier of flowering on most genotypes (Figure 2). Of the 19 genotypes evaluated, 16 had an increase in the number of seeds produced, with 8 being significant at the 90% confidence level. 

As findings were obtained, field application methods involving timing and dose were slightly adjusted each year to improve treatment effectiveness across this wide diversity of genotypes, as outlined in Table 3. The three application methods, M1, M2 and M3, all significantly increased the number of seeds produced compared to the control (Table 3). 

M1 treatments were applied uniformly to all plants when they were 6 weeks old and were continued using a set schedule for the course of the growing season. Excessive phytodamage due to STS was seen after 5 weeks of treatment; treated plants had an average damage rating of 2.57, and 10 percent were killed due to excessive PGR applications. To avoid excessive damage, M2 treatments commenced when a given plant was over 60 cm tall and had already forked. Additionally, plants were assessed weekly for damage using a 0–3 rating scale (Appendix A). If an individual plant had a rating of 2, concentrations of PGR were reduced by half. If symptoms were 3 (most severe), PGR applications were skipped until the evaluation on newly expanded leaves was lower than 3 in subsequent weeks. This adjustment in the M2 method reduced the damage level to an average damage rating of 0.45, and the number of plants killed did not differ from the controls. The M2 method largely avoided damage; however, as plants grew larger, the dosage was apparently insufficient to elicit the desired effect, and the average increase in seed number relative to controls was only 0.43 per plant at tier 1. The M3 protocol was modified to increase the dose of STS by 1 mL for every 20 cm of height over 60 cm. This modification successfully kept phytodamage to an acceptable damage rating of 1.22, and the number of plants killed was insignificant, while it increased the seed production at tier 1 to 2.65 per plant. 

The M3 method, which employed all of the modifications based on experience with M1 and M2, was evaluated for whole-season flower and seed production on whole plants encompassing the first pruned tier in the plant growth regulator and pruning treatments and up to four tiers in the control treatment (Figure 3). The mean female flower counts and seed counts were substantially greater for M3 compared to the control. These gains were obtained despite the limited number of tiers present in pruned plants where no further branching events and tiers are formed after fork branches are removed. 

## 3. Discussion

### 3.1. Greenhouse Elucidation of Treatments

The current investigations indicated that all three treatment components, STS, BA and pruning, were individually beneficial in enhancing flower and seed development. Furthermore, the component treatments were additive, as the greatest increase in flower production was when they were used in combination (Figure 1, Experiment 1 and Experiment 4). These findings advance our knowledge of approaches that can be used to obtain flowers and seeds in cassava breeding programs. Previously published field studies aimed at enhancing flowering and seed production with PGRs and pruning have provided encouraging results; however, the magnitude of benefit has been inconsistent. Some of these studies were limited to just one PGR, either BA or STS [15,22], or STS was applied as a spray rather than via petiole influx [24]; furthermore, pruning was not included in these studies [22,24]. Most encouraging have been studies that included the full set of treatments as described in the present investigation, where benefit was obtained for both flower numbers and the proportion of female flowers [23,25]. However, these studies did not fully determine the effects of BA on feminization and STS on preventing flower abortion and flower longevity, and they did not assess seed sets. Furthermore, these studies did not address inconsistencies due to suboptimal timing of treatment application when a genotype’s first flowering is unusually early or late, in which phytodamage occurred in some genotypes. As discussed below, the current field studies addressed these additional issues.

Our studies confirm and substantiate the findings of Oluwasanya et al. [23] that STS can be effectively applied to cassava through a cut petiole using the negative water pressure (tension) in the plant xylem to draw the solution into the vasculature such that the STS is then redistributed to the apical meristem. Lin et al. [26] demonstrated the effectiveness of petiole feeding and xylem transport through experiments using soybean plants fed with a range of aqueous solutions containing tracer dyes, small metabolites and radiolabeled chemicals. Similarly, studies with Ag nanoparticles showed that whereas foliar and root uptake and transport were limited, petiole feeding and trunk injection delivered a large amount of nanoparticles via xylem into plants [27]. Studies of STS applications in cassava showed that when applied to older, mature leaves, silver was taken up and transported to the young leaves and stem tissues of the apical region where inflorescences develop [22]. While the Ag^+^ ion is relatively immobile due to the positively charged ion binding to the anionic surfaces of the xylem, its mobility is enhanced when complexed with thiosulfate [28]. Furthermore, observations by Hyde et al. [22] suggested that a benefit of petiole application is that with this method, silver is delivered to young tissues with less phytodamage compared to spraying to sensitive young apical tissues, which incurred more damage. In contrast to the effectiveness of petiole feeding of STS via xylem, BA feeding through the petiole was not effective (Figure 1, Experiment 3). For BA, an increase in flower numbers was obtained when it was applied as a spray to the apical region, especially when applied in combination with petiole-fed STS (Figure 1, Experiments 3 and 4). 

In greenhouse experiments, the benefit from applied STS plateaued at 0.25 mM, and no further benefit was seen beyond this concentration (Figure 1, Experiment 2). STS and BA performed in a similar manner in the field as in the greenhouse; however, a considerably higher concentration of STS was needed to elicit a similar benefit. This was seen by comparing Experiment 1 in the greenhouse and the Ibadan field experiment. Both experiments used the same methodology; however, the trial grown in Ibadan used concentrations four times higher than those used in the greenhouse. 

Previous studies on the mechanisms of action of STS have indicated that silver binds and thereby blocks ethylene receptors, preventing ethylene response signal transduction [28,29]. Studies have identified numerous roles for ethylene in fruit ripening [30], fruit abscission [31], hastening flower senescence [29] and arresting inflorescence development and flower formation [22]. Studies in cassava demonstrated that when plants were pre-treated with the ethylene-generating compound ethephon, STS blocked flowering arrest and flower abscission [22]. Transcriptomic studies on cassava indicated that enhancement of flower development by STS + BA was accompanied by downregulation of several genes associated with repression of flowering, creating widespread changes in the network of hormone signaling by ethylene and other regulatory factors [23].

The current greenhouse studies provide evidence for the role of the cytokinin BA in feminizing flower development, as reported in previous studies of cassava [23,25,32] and in *Jatropha curcas* L., which, like cassava, is in the family Euphorbiaceae [33], where the cytokinin Forchlorfenuron was effective. Indeed, in the present investigation, the percentage of female flowers increased from about 6–23% in STS-only treatments to over 89% in treatments that included BA (Table 1). Furthermore, recent studies on Guinea yam showed that pruning and STS were effective in increasing the number of spikes per plant and the flowering intensity on both sparse flowering and monoecious cultivars [34]. Hence, for breeding programs where the goal is the production of large numbers of seed progeny from desired parents, there is considerable potential benefit from the enhancement of female flower production using the combination of treatments described here.

### 3.2. Field Performance of PGR-P Methods

As the present work advanced from initial greenhouse studies to field trials, we found that considerably higher concentrations of STS were required in the field compared to what was effective in the greenhouse [23]. Thus, for method M1, we boosted the concentration from 0.5 to 2.5 mM STS. We also found in preliminary work that BA and STS treatments had to be started about 2 weeks before fork-type branching, an early indicator of flower initiation, to be effective. To ensure that early-flowering lines would receive PGRs early enough to be effective, we began treatments 5 weeks after planting. However, M1 resulted in a high average damage rating, and a high percentage of plants were killed (Table 1). 

Phytodamage from high STS applications has long been recognized [17,29], and in field trials with cassava exposed to PGR treatments similar to those in the present study, authors describe phytodamage as a problem that limits treatment benefits [23,32]. Small young plants were particularly susceptible to BA and STS damage when applied at high concentrations. Nevertheless, our findings indicated that to enhance seed production, dosage needed to be in a range where some foliar damage was observed (Appendix A). Furthermore, as plants grew larger at later stages, applied STS became diluted into larger biomass such that its effectiveness diminished. To achieve an optimal dosage, in M3, we increased dosage in proportion to plant height (Table 1). Method M3 successfully kept the average damage rating at an acceptable score of 1.22, while the number of seeds produced per plant was more than double the number of seeds produced without the treatments (Figure 3). 

Our goal has been to identify methods that provide tools for breeding programs to reliably and rapidly obtain progeny from a broad range of genotypes. In addition to producing more seed, obtaining seed at earlier tiers facilitates speed breeding, which is necessary for breeding methods with rapid one-year cycles [5,32]. However, our analysis does not evaluate the time and effort needed to monitor plants and apply treatments. Hence, we expect that breeders might find that it is possible to achieve their needs for progeny production with a more limited treatment regimen using only one or two treatment components. For some valuable but recalcitrant genotypes, the full set of treatment components will be needed. For example, Rodrmguez et al. [32] pointed out that the CIAT cassava breeding program needs cultivars with erect, non-branching plant architecture to allow farmers to use them in high-density plantings. Such genotypes are good candidates for PGR and pruning methods because they are late flowering and usually do not generate many seeds in crossing nurseries. Furthermore, there are environmental effects on flower timing and development [14,17,21,35], and different locations might have different optimal treatment strategies. The present investigation developed a protocol that adjusts the dosage and timing of treatments to accommodate differences in growth, time of flowering and sensitivity to the growth regulators so that flower and seed production are enhanced among a wide range of environments and genotypes and cassava breeding programs can successfully use a diversity of germplasm to develop cultivars with needed traits. 

## 4. Materials and Methods

### 4.1. Plant Materials

#### 4.1.1. Greenhouse Experiments

Three cassava genotypes, standard reference lines in Africa, were used in the greenhouse experiments. TMS-IBA-980002 (also known as TMSI980002) and TMEB419, provided by the International Institute of Tropical Agriculture (IITA), Ibadan, Nigeria, and Nase14, provided by the National Crop Resources Research Institute (NaCRRI), Namulonge, Uganda.

#### 4.1.2. Ibadan Experiments

Three cassava genotypes, standard reference lines that are used in Nigerian breeding programs, were used in Ibadan experiments: TMS-IBA-980002 (also known as TMSI980002) and TMEB419 and TMS30572 (also known as IITA-TMS-IBA30572, TMSI30572 and Nase 3).

#### 4.1.3. Ubiaja Experiments

The genotypes CR36_5, F10_30_R2, IITA_TMS_IBA160008, KS12000196, MKUMBA, TMEB2, TMEB412, TMEB419, TMEB693, TMEB7, TMS14F1313P0076, TMS15F1463P0008, TMS15F1463P0054, TMS15F1482P0051, TMS15F1482P0098, TMS17F2026P0001, TMS17F2028P0011, TZ_130 and UCC_20001212 from the IITA breeding program, known to produce low levels of flowers and seed, even when grown in Ubiaja, a location known for enhancing flower development, were used for the evaluation and development of flower enhancing methods in 2019, 2020 and 2021 [23].

### 4.2. Growth Conditions

#### 4.2.1. Greenhouse

Stem cuttings (stakes) were cut to 15 cm length from the bottom 1 m of previously grown plants. Stakes were planted into 11 L pots (Polytainer #3; Nursery Supplies Inc., Chambersburg, PA, USA). Rooting media was a mixture of peat:vermiculite:perlite (62:22:11; *v*:*v*) with added dolomitic limestone and 2.2% (*w*:*v*) of fertilizer (10-5-10 Jacks Pro Media mix plus III; J.R. Peters, Inc., Allentown, PA, USA), as previously described [22]. Plants were grown in a greenhouse at Cornell University, Ithaca, NY, USA, with supplemental heating as needed to attain a temperature of 30 °C (day) and 25 °C (night). Supplemental lighting from 400 Watt metal halide lamps spaced at 80 × 190 cm (PX-MPS400/7 K, Plant-Max, 1000Bulbs Co., Garland, TX, USA) was provided between 06:00 a.m. and 20:00 p.m. when solar photosynthetic (400–700 nm) photon flux density was <500 μmol m^−2^ s^−1^.

#### 4.2.2. Ubiaja Location

Located on the outskirts of Ubiaja, Nigeria, at 6.6728 N, 6.3557 E, with an elevation of 220m, the field was cultivated, ridged and marked for 1 m × 1 m plant spacing. Genotypes were planted with stem cuttings of approximately 20 cm in length. Fields were fallow the previous year. The land was tilled and ridged with no extra nutrients or soil amendments added. Fields were kept free of weeds with hand weeding. The temperatures in shielded, naturally ventilated weather shelters were logged at 15 min intervals (see supporting data at: https://cassavabase.org/ftp/manuscripts/Hyde_et_al_2023/ (accessed on 26 January 2024)).

#### 4.2.3. Ibadan Location

Experiments were conducted at the International Institute of Tropical Agriculture (IITA), Ibadan, Nigeria 7.4944 N, 3.8974 E. The field was cultivated, ridged and marked for a 1 m plant spacing. Genotypes were planted with stem cuttings of approximately 20 cm in length. 

### 4.3. Treatment Applications

#### 4.3.1. STS and BA Spray

Spray application of PGRs was performed by applying a fine mist of the solution with a hand pump sprayer to the apical region of each branch weekly. Spraying coated the desired leaves or inflorescence until runoff, usually requiring about 3 mL. Before flowering, spraying was targeted to the 3 to 5 youngest immature (folded) leaves; as the inflorescence developed, spraying was directed onto the inflorescence. 

#### 4.3.2. STS Petiole Infusion

The desired STS solution was put into a 15 mL plastic tube. A leaf and petiole 40–50 cm below the apical meristem was used. The petiole was submerged in a shallow dish of water and cut under water with surgical scissors to remove the leaf blade. To prevent air lock of the xylem, the cut end of the petiole was kept submerged in the water for a few seconds until it was quickly transferred and submerged into the STS solution (described below for each experiment). Care was taken to avoid kinking the petiole, which may collapse the xylem vessels.

#### 4.3.3. Pruning

Plants were inspected 2–3 times a week to identify newly initiated inflorescences and fork-type branching. Newly formed branches were removed by pinching off with fine forceps as soon as they were visually identified, approximately 3–5 mm in length. 

### 4.4. Data Collection

Flower buds > 2 mm in diameter were counted weekly. Non-senescent flowers that reached anthesis were counted and designated as male or female. Plants were evaluated weekly using a 0–3 damage rating scale (Appendix A), with 3 indicating severe damage. 

### 4.5. Experimental Designs

#### 4.5.1. Greenhouse Experiment 1: STS Sprayed + BA Sprayed

Two batches of plants were grown under greenhouse conditions 8 months apart in a randomized complete block design (RCBD). Each batch contained four replicate blocks of three genotypes (TMSI980002, Nase14 and TMEB419). The four treatments evaluated were: control (sprayed with water), STS (0.125 mM) sprayed, BA (0.5 mM) sprayed and STS (0.125 mM) plus BA (0.5 mM) sprayed together.

#### 4.5.2. Greenhouse Experiment 2: STS via Petiole

Four replicate blocks of the genotypes (TMSI980002, Nase14 TMEB419) were arranged in an RCBD in the greenhouse. Two applications of 2.5 mL of STS were applied via the petiole at five treatment concentrations (0, 0.125, 0.25, 0.5 and 1.0 mM) two weeks apart prior to floral initiation. 

#### 4.5.3. Greenhouse Experiment 3: STS and BA Spray, Petiole Combinations

Four replicate blocks of the genotypes (TMSI980002, Nase14 and TMEB419) were arranged in an RCBD. Five treatments were applied: (a) 2.5 mL of 0.5 mM STS plus 0.5 mM BA infused together via petiole; (b) 2.5 mL of 0.5 mM STS via petiole plus 0.5 mM BA spray; (c) 0.125 mM STS spray plus 0.5 mM BA spray; (d) 0.125 mM STS spray plus 2.5 mL of 0.5 mM BA infused via petiole; and (e) control (sprayed with water). 

#### 4.5.4. Greenhouse Experiment 4: STS and BA with Pruning

Four replicate blocks of the genotypes (TMSI980002, Nase14 and TMEB419) were arranged in an RCBD in the greenhouse. Five treatments were applied, including control, pruned, pruned with BA (0.5 mM spray), pruned with STS (0.5 mM petiole) and pruned with BA (0.5 mM spray) plus STS (0.5 mM petiole). 

#### 4.5.5. Ibadan Field Trial

Treatments of STS (0.5 mM), BA (0.44 mM) and both STS and BA were applied to the genotypes I30572, TMSI980002 and TMEB419 weekly until plants exhibited phytodamage and compared with an untreated control. Four replications of each genotype treatment combination were randomly assigned and evaluated. 

#### 4.5.6. Ubiaja 2019–2021 Field Trials

The combination of pruning, STS and BA was evaluated with a set of nineteen genotypes over the course of 3 years in an RCBD. The trial in 2019 had 3 replicates, and trials in 2020 and 2021 had 8 replicates each. Three plant growth regulator and pruning (PGR-P) treatment methods (M1, M2 and M3) were used, one each year. Allowing for timing and dose adjustments to optimize the treatment package for a wide range of phenotypes. For M1, STS (4.0 mM) and BA (0.5 mM) treatments were started at 5 weeks of age on all plants and continued every week (for BA) or every other week (for STS). Pruning was conducted on the first branching event (first tier). For M2, STS (4.0 mM) and BA (0.5 mM), treatments were started when the plants were 60 cm tall, and the concentration of the treatments was reduced based on assessed damage. Pruning in 2020 was applied to the branching event, which happened ≥one week after the first STS treatment. The treatments for M3 were similar to those in 2020; however, the dose of STS was increased by 1 mL for every 20 cm of additional height above 60 cm. In all three methods, female flowers were manually pollinated for 7 days starting at anthesis. 

### 4.6. Statistical Analysis 

Mixed-model ANOVA was used, which included the fixed effects of treatment, genotype and genotype by treatment interaction. Batches (repetitions at different times), which accounted for batch-to-batch variation, and replicate blocks of plants were modeled as random effects. Greenhouse blocks were assigned based on uniform height, and field blocks were assigned based on location in the field. The lm and ANOVA functions of the “stats” package conducted in R studio (v2023.12.0; R version 4.0.3) were used to model the effects [36]. The “emmeans” package (v1.7.5) [37] was used for mean comparisons both pairwise with *t*-tests and with multiple tests using Tukey–Kramer honest significant difference tests.

## 5. Conclusions

In this study, we elucidated, systematically developed and tested a robust protocol aimed at augmenting seed production in cassava-crossing nurseries. Our findings demonstrate the significant impact of anti-ethylene, cytokinin and pruning treatments, both individually and in combination, on enhancing flowering and seed yield in cassava. Specifically, the administration of STS via petiole infusion and BA application as an apical spray, coupled with young branch pruning, resulted in a significant increase in flower numbers. Notably, greenhouse trials showed that the combination of pruning, BA and STS yielded an average of 168 flowers per plant at the first tier of flowering. While STS primarily augmented flower numbers, BA significantly increased the proportion of female flowers, thereby enhancing the production of progeny from desired parent plants. Furthermore, refinement of our protocol through a three-year implementation on a set of 19 genotypes in practical field conditions maintained acceptable phytodamage levels while substantially elevating seed production per plant compared to untreated plants. This protocol’s adaptability, allowing for dosage and timing adjustments to accommodate genotypic and environmental variations, equips cassava breeding programs with powerful tools with which to utilize a diverse range of germplasm. 

## Figures and Tables

**Figure 1 plants-13-00382-f001:**
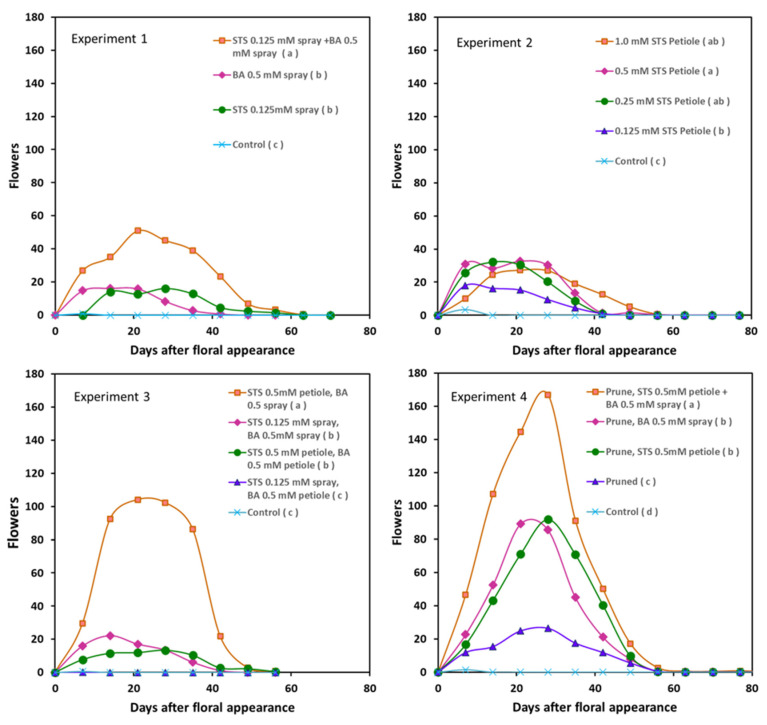
The effect of benzyladenine (BA), silver thiosulfate (STS) and pruning in various combinations on the number of flower buds and non-senescent flowers per inflorescence. Shown are the averages of the genotypes TMSI980002, NASE14, TMEB 419 across four biological replicates. Symbol legends indicate (in parenthesis) statistical treatment comparisons; treatments that do not have the same letter are significantly (*p* ≤ 0.05) different using the Tukey HSD multiple range test to evaluate the flower integral. See Section 4 section for details. Comprehensive statistical analysis of the maximum, retention duration and integral (area under the curve) of flower counts for each genotype is included in Appendix A.

**Figure 2 plants-13-00382-f002:**
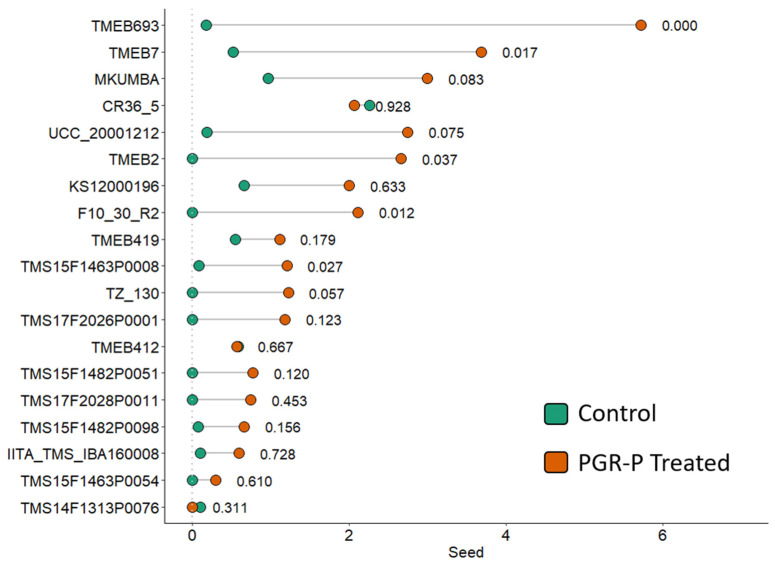
Seed produced per plant at the first tier of plants receiving control treatment vs. those treated with plant growth regulator and pruning (PGR-P) treatments in 19 genotypes over three years in Ubiaja. The *p*-values following the line segments indicate the probability that treatment differences from controls can be explained by chance alone.

**Figure 3 plants-13-00382-f003:**
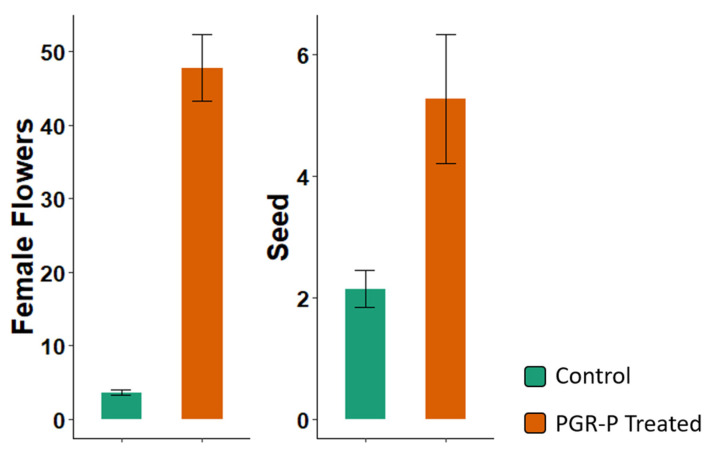
Flowers and seeds produced per plant in plants receiving control treatment vs. those treated with plant growth regulator and pruning (PGR-P) treatments using method M3. Shown are the averages ± SEM on the entire plant up to tier 4 in 19 genotypes over three years in Ubiaja.

**Table 1 plants-13-00382-t001:** The effect of benzyladenine (BA) and silver thiosulfate (STS) on the percent of female flowers. Treatment concentrations are as indicated in Figure 1.

Experiment 1			Experiment 2			Experiment 3			Experiment 4		
Treatment	Percent Female		STS Treatment	Percent Female		Treatment	Percent Female		Treatment	Percent Female	
Control	0	a *	0 mM	0	a	Control	0	a	Control	0	a
STS	18	a	0.125 mM	17	a	STS Spray, BA Petiole	0	a	Prune	6	a
BA	100	b	0.25 mM	6	a	STS Petiole, BA Petiole	65	b	Prune, STS	15	a
STS+BA	91	b	0.5 mM	13	a	STS Spray, BA Spray	89	b	Prune, BA	98	b
			1.0 mM	23	a	STS Petiole, BA Spray	100	b	Prune, STS + BA	87	b

* Comparisons between treatments that do not have the same letter are significantly (*p* ≤ 0.05) different using the Tukey HSD multiple range test.

**Table 2 plants-13-00382-t002:** Effect of BA and STS treatments on flowering and fruit-set traits in a field trial in Ibadan, Nigeria. Treatments (0.44 mM BA, 0.5 mM STS) were applied as a spray to the apical region. Shown are the averages of four replicates of three genotypes, I30572, I980002 and TMEB419.

Treatment		Days to Flowering	Female Flowers	Male Flowers	Flower Integral	Fruit Set
Control		67	0	0	1	0
BA		59	1	5	20	0
STS		73	0	8	32	0
STS + BA		65	10	33	133	5
Genotype	Pr (>F) ^†^	***	NS	NS	NS	NS
PGR treatment		NS	*	***	***	*
G × T		NS	NS	NS	•	NS
pooled SEM		4.7	3.3	7.4	26.6	2.0

^†^ Symbols indicate the probability that an F value could be attributed to chance: *** is a *p*-value < 0.001; * is a *p*-value < 0.05; • is a *p*-value < 0.10; NS is not significant based on an ANOVA model of treatment, genotype and treatment X genotype.

**Table 3 plants-13-00382-t003:** Effects of three plant growth regulator plus pruning (PGR-P) treatment protocols (M1, M2 and M2) relative to controls on seed numbers, phytodamage and percent of plants killed in Ubiaja, Nigeria. Values indicated are the differences between treatment and control. Values for control treatments in trials of M1, M2 and M3, respectively, were as follows: numbers of seed/plant were 0.61, 3.44 and 2.43; damage ratings were 0.0, 0.0 and 0.0; percent of plants that did not survive were 3.5, 0.3 and 0.28.

Method *	PGR Start	Dose	PGR Frequency	Start of Pruning	Tiers Treated	Seed Increase ^†^	Damage Increase	Percent Killed
M1	5 weeks after planting	2.5 mL 4 mM STS, 0.5mM BA	STS biweekly, BA weekly	At first tier	Tier 1	2.57 *	2.15 ***	10.6 ***
M2	At 60 cm height	M1, but reduce dose if plants are moderately damaged	M1 but delay if plants are heavily damaged	At the tier after 60 cm in height	Tier 1 and 2	0.43 **	0.45 ***	0.5 NS
M3	At 60 cm height	M2, and increase dose with height	M1 but delay if plants are heavily damaged	At next tier, ≥1 week after first PGR application	Tier 1–4	2.65 ***	1.22 ***	0.3 NS

* A full description of each method is outlined in Section 4; a detailed standard operating procedure (SOP) for M3 is in the supporting data website. ^†^ The increase in seed produced per plant, damage and percent killed between control and each application method. Symbols indicate the probability a difference could be attributed to chance: *** is a *p*-value < 0.001; ** is a *p*-value < 0.01; * is a *p*-value < 0.05; NS is not significant based on an ANOVA model of treatment, genotype, block and treatment X genotype.

## Data Availability

Data supporting the reported results presented in this study are publicly available. This data can be found here: https://cassavabase.org/ftp/manuscripts/Hyde_et_al_2023/ (accessed on 26 January 2024).

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
