# Peer review of "Development of Methods for Improving Flowering and Seed Set of Diverse Germplasm in Cassava Breeding"

_plants, 2024, doi:10.3390/plants13030382_

Round 1
Reviewer 1 Report
Comments and Suggestions for Authors
· Give more detail about suitable treatments in the abstract.
Results
· Make Table 1 like Figure 1.
Materials and methods
· Give more details about plant materials used, data collection, and field trails in all experiments.
Author Response
Reviewer 1 (round 1)
Note: all line numbers refer to the track changes version of the document
- “Give more detail about suitable treatments in the abstract”
We agree that more details would strengthen the abstract and have added material in L17-22
- “Make Table 1 like Figure 1.”
Figure 1 presents flowering data on a day-to-day basis, which is beneficial in showing both the flower counts and the longevity of flowers. We appreciate that the reviewer recognizes the value of this presentation. However, this level of time-dependent detail becomes less informative when we plot percent female because female flowers develop earlier and end sooner than males, such that the % female data reflect the asynchrony rather than the feminization phenomenon. Our assessment is that the single average value of %female is more informative. Moreover, we have provided the full set of raw data in the set of supporting data at the web site referenced on L513-514, so readers can examine alternative ways of analyzing the data. We have therefore retained the table form of presentation in Table 1.
- “Give more details about plant materials used, data collection, and field trails in all experiments.”
We agree more details on M&M are needed, and therefore we have added them on plant materials (L334, L341), growth conditions (L362-367, 373-376), and treatments (L428-432).
Reviewer 2 Report
Comments and Suggestions for Authors
The authors have made great efforts to develop a useful protocol to improve the flowering and seed set of some cassava germplasm for the breeding program. Generally, the obtained results are of interest to the readers, especially to Agronomists and Breeders. However, some weak points have been found in the current form of the manuscript, which should be made in order to improve the quality of the study (further see some comments in the pdf file).
1. The title may be changed because the current study included some experiments conducted in the greenhouse condition;
2. The abstract must be improved, and should add the most significant findings of this study;
3. Some minor revisions should be made in the introduction part (see in pdf file)
4. Materials and Methods should be judiciously rechecked and added some specific information.
5. Why did the authors select those three cassava genotypes for the greenhouse experiments and Ibadan experiments? while Ubiaja experiments used some different genotypes?
6. No information on the flowering times of those genotypes was narrated or discussed in the manuscript;
+Greenhouse conditions should add some specific information such as temperature, humidity, light, watering etc, which may affect the attained results
7. Field conditions in both Ubiaja and Ibadan must add some specific conditions including areas height, temperatures, lighting time etc.
8. The discussion part should give some hypotheses as to why STS, BA and pruning significantly augmented the number of flowers.
9. Conclusion should be added

Comments on the Quality of English LanguageDear Editor
As reviewed above, in my opinion, it is a good work and will be widely interesting for readers and a useful tool for the cassava breeding program. However, it needs some major revisions as listed above
Best regards
TD Khanh
Author Response
Reviewer 2 (round 1)
Note: all line numbers refer to the track changes version of the document
- “The title may be changed because the current study included some experiments conducted in the greenhouse condition”
Thank you for this suggestion. We agree and have omitted "practical field" from the title
- “The abstract must be improved, and should add the most significant findings of this study”
We have added more detail to the Abstract with the most significant findings (L17-22)
- “Some minor revisions should be made in the introduction part (see in pdf file)”
We have reviewed all suggestions and have made changes to introduction (and other sections) as shown on reviewer's pdf. Thank you.
- “Materials and Methods should be judiciously rechecked and added some specific information”
We have carefully checked the M&M section and made several revisions.
- “Why did the authors select those three cassava genotypes for the greenhouse experiments and Ibadan experiments? while Ubiaja experiments used some different genotypes?”
The greenhouse (Figure 1) and Ibadan experiments (Table 2) were studies that used standard reference lines available at both Cornell and Nigeria. The Ubiaja study targeted a broader range of lines that are actively used in the IITA cassava breeding program intended to represent a set of genotypes with a range of flowering behavior for which the treatments would be used in the future. We have added explanatory text on L83, L136, L163, L334.
- “No information on the flowering times of those genotypes was narrated or discussed in the manuscript”
Supplementary Tables S1 to S4 contain data on flowering times. We have added a sentence to the Results section to describe these findings (L89), and referenced these tables.
- “Field conditions in both Ubiaja and Ibadan must add some specific conditions including areas height, temperatures, lighting time etc.“
We have added information on land preparation and reference to temperature and light data which was logged at the site. L373-376
“+Greenhouse conditions should add some specific information such as temperature, humidity, light, watering etc, which may affect the attained results “
We have added information to the M&M which provides greenhouse conditions (L362-367)
- “The discussion part should give some hypotheses as to why STS, BA and pruning significantly augmented the number of flowers.”
Thank you for this valuable suggestion. We have added some sentences which review the mechanisms of action of the treatments (L241-248; L260-270).
- “Conclusion should be added”
We have added a conclusions section after the M&M (L474-491)
Reviewer 3 Report
Comments and Suggestions for Authors
It is not necessary to add “described in M&M” to refer to the method section in scientific papers.
After an abbreviations is introduced, only use the abbreviation. E.g. STS and BA are mentioned twice.
Choose a more detailed headline instead of 2.1. Greenhouse and 2.2. Field trials.
Line 126: Did you only use STS? Because Table 2 lists BA and STS.
Table 2: add full name of treatment abbreviations in the legend.
Discussion line 243-258 is a repetition of the results section and not a discussion. Change accordingly.
Line 252: change “a slight amount” to a more scientific term.
Line 273-277: change to include that you have achieved the aim of the study and point out the contribution of your results to the scientific community.
Author Response
Reviewer 3 (round 1)
Note: all line numbers refer to the track changes version of the document
- “It is not necessary to add “described in M&M” to refer to the method section in scientific papers”
We have removed unnecessary statements of this sort from line 93, 103, 183
- “After an abbreviations is introduced, only use the abbreviation. E.g. STS and BA are mentioned twice”
We have removed secondary definitions of the abbreviations in the Results. We note in the Plants Instructions to Authors that "Abbreviations should be defined in parentheses the first time they appear in the abstract, main text, and in figure or table captions and used consistently thereafter." Our understanding of this is that we should have definitions for abbreviations the first time they were mentioned in figure or table legends (in addition to Abstract and first time used in the main text). Of course we defer to the Style Editor if we have misunderstood.
- “Choose a more detailed headline instead of 2.1. Greenhouse and 2.2. Field trials “
We have provided more detailed headlines.
- “Line 126: Did you only use STS? Because Table 2 lists BA and STS.”
Good point. We have revised to clarify (rearranging the sentences in the process). (L132-136)
- “Table 2: add full name of treatment abbreviations in the legend”
We have added definitions for treatment abbreviations the first time they appear in the abstract, main text, and in figure or table captions, as explained above (L84, L112, L127)
- “Discussion line 243-258 is a repetition of the results section and not a discussion. Change accordingly”
We agree and have omitted the repetitious material (L296-303)
- “Line 252: change “a slight amount” to a more scientific term.”
We have changed from "slight amount of" to "some"
- “Line 273-277: change to include that you have achieved the aim of the study and point out the contribution of your results to the scientific community.”
Excellent suggestion. We have modified accordingly. (L325-329)